# Influence of Homogenization Time and Speed on Rheological and Volatile Composition in Olive-Based Pâtés

**DOI:** 10.3390/foods8040115

**Published:** 2019-04-04

**Authors:** Francesco Caponio, Graziana Difonzo, Giacomo Squeo, Stefania Fortunato, Roccangelo Silletti, Carmine Summo, Vito M. Paradiso, Antonella Pasqualone

**Affiliations:** Department of Soil, Plant and Food Science (DISSPA), University of Bari Aldo Moro, Via Amendola, 165/a, I-70126 Bari, Italy; graziana.difonzo@uniba.it (G.D.); giacomo.squeo@uniba.it (G.S.); stefania.fortunato@uniba.it (S.F.); s.roccangelo@libero.it (R.S.); carmine.summo@uniba.it (C.S.); vito.paradiso@uniba.it (V.M.P.); antonella.pasqualone@uniba.it (A.P.)

**Keywords:** pâtés, olive, rheology, texture analysis, syneresis, volatile compounds

## Abstract

The influence of the homogenization time and speed on rheological and volatile composition in olive-based pâtés was studied. Five experimental trials were performed applying different combinations of time and speed homogenization: 1, 3, and 5 min at 12,000 rpm and 4000, 8000, and 12,000 rpm at 5 min. The obtained results showed that the processing parameters of the homogenization step significantly influenced the rheological and sensory properties of olive-based pâtés. Both time and speed influenced the rheological properties of the product. The increase of homogenization time and speed determined a significant reduction of hardness and syneresis. As regards color indices, significantly higher *L** values were obtained when intermediate time and speed conditions were applied, whereas *a** and *b** indices showed a not univocal behavior. Both time and speed variables also influenced the volatile fraction of the pâtés (higher homogenization speed and time corresponded to higher terpenes and aldehydes).

## 1. Introduction 

Pâtés are processed products having an important gastronomic tradition, characterized by good sensory properties with a rough texture [1]. The main ingredients of pâtés are finely minced and mixed with various secondary ingredients considered essential for their binding capacity. These products are usually consumed as either seasonings for pasta or dressings for meat, salads, and sandwiches [2]. Pâtés are able to satisfy the consumer’s growing demand since they represent value-added products with good flavor and adequate shelf-life. 

In recent years several studies have focused on these products. In particular, Marudova et al. [3] described the thermal characteristics of poultry pâtés which were enriched with vegetable ingredients in relation to their chemical composition and technological process. Guimarães et al. [4], investigated the application of the residue from soy drink and tofu in the development of vegetable paste formulations in order to obtain a product with good acceptability and nutritional quality. Cosmai et al. [5], evaluated the effects of freezing storage on the oxidative stability, bio-active compounds and color characteristics of two different type of non-thermally treated tomato-based pâtés compared to the same pâtés subjected to thermal stabilization and stored at room temperature. Cosmai et al. [6], with the aim to minimize the effect of thermal treatment, analyzed the effect of the combination of a natural extract and modified atmosphere packaging on the shelf life extension of fresh olive-based paste. Moreover, the development of new formulations of olive-based pâtés was investigated by Cosmai et al. [7] to satisfy more dynamic, complex, and differentiated consumer demands for traditional and functional foods. Gorlov et al. [8] developed a technology to obtain functional meat-vegetable pâtés based on mutton and poultry by-products with addition of chickpeas. Furthermore, the effects of substituting pork back-fat with different levels of sunflower and canola oil combinations in chicken liver pâtés were studied (Xiong et al. [9]); the authors further evaluated the effects on pre-emulsification back-fat and oil on the physicochemical properties, fatty acids profile, microstructure, textural, and sensory properties of fine spreadable chicken liver pâtés. 

Among all food sensory attributes, visual (freshness, color, defects, and decay), textural (crispness, turgidity, firmness, toughness, and tissue integrity), and flavor (taste and smell) characteristics play main roles in consumer purchase behavior. For vegetables-based pâtés, the production process consists, in general, of the following steps: Selection of the raw materials; preliminary operations (washing, cleaning, pealing and destoning); blanching; shredding and homogenization of ingredients; filling and seaming; thermal treatment and cooling. 

Among these steps, the homogenization operation represents a crucial point which influences the rheological characteristics (texture and syneresis) of the product [10,11,12,13,14], as well as its flavor. In addition, the homogenization step might induce the enzymatic depletion of antioxidants, as a result of cellular disruption which allows contacts of these substrates with oxygen and enzymes influencing the shelf-life [15]. Lopez-Sanchez et al. [16] and Colle et al. [17] investigated the influence of high-pressure homogenization on the rheological and microstructural characteristics of vegetable pâté, as well as on the release of bioactive compounds and antioxidant activity.

Pâtés have been produced for a long time with the aim to extend the shelf-life of vegetables. In the Mediterranean basin, which holds the major production of olives worldwide, the olive processing as pâtés is a consolidated practice since many are not suitable for table olives production due to their size and/or shape. Decades ago, this practice was mostly homemade, while today most of olive-based pâtés are produced industrially. From a nutritional point of view, olives are rich in monounsaturated fatty acids (oleic) and bioactive compounds, such as polyphenols, considered to be useful for human health [18]. These characteristics make olive pâtés not only attractive from a sensorial and commercial point of view but also from a nutritional one.

To the best of our knowledge, no investigation has been carried out on the influence of the homogenization step of olive-based pâtés on their rheological properties and volatile composition. As such, the present investigation was aimed at studying the effect of two fundamental processing parameters of the homogenization step, time and speed, on the overall physico-chemical properties of the final mixed system. 

## 2. Materials and Methods

### 2.1. Sample Preparation

The olive-based pâtés (OBP) were produced using table olives (770 g kg^−1^; *Olea europaea* L., cv. *Bella di Cerignola*), debittered by the Spanish method, extra-virgin olive oil (150 g kg^−1^; cv. *Coratina*), salted anchovies (40 g kg^−1^, *Engraulis encrasicolus* L.), red onion (40 g kg^−1^, *Allium cepa* L.), arugula (*Eruca vesicaria* (L.) Cav.), and a few drops of balsamic vinegar. All ingredients were purchased at local retailers (Bari, Apulia, Italy).

Before the production of olive-based pâtés, raw materials were subjected to preliminary treatments as follow. Table olives were washed with tap water and destoned. Blanching in boiling water was carried out for red onion and arugula. All these treatments were performed using a pilot plant to simulate the industrial production of this type of product. Each ingredient was gradually added and mixed (about 500 g) in a 1 L stainless steel homogenizer (Waring LB 20 E S, Rome, Italy). 

Five experimental trials were performed applying different combinations of time and speed homogenization. In particular, homogenization was carried out at a constant speed of 12,000 rpm for 1, 3, and 5 min, whereas, three different speeds (4000, 8000, and 12,000 rpm) were applied for a constant time of 5 min, as summarized in Table 1. After mixing, the samples were immediately analyzed. Three independent replications were carried out for each trial.

### 2.2. Rheological Analysis 

Sample hardness was measured with a Texture Analyser (Z1.0 TN, Zwick GmbH & Co. KG, Ulm, Germany), a 1000 N load-cell, and the software, Text Expert 2 (Zwick-Roell, Kennessaw, GA, USA). The analysis was conducted at room temperature (23 °C) and consisted of a 2-cycle compression using a 10 mm diameter cylindrical probe. The samples were previously conditioned for the time needed to reach the proper temperature. A 50 mL beaker (with a diameter of 45 mm and a height of 60 mm, Duran, Meinz, Germany) was filled with pâté and the surface was smoothed with a palette knife. The pâté filled the beaker to a depth of 50 mm. Afterwards, the plunger penetrated to a depth of 30 mm from the surface into the sample at a constant speed of 150 mm min^−1^. Force-time deformation curves were recorded, and the maximum force applied was recorded as the hardness (N) of the sample. Values were given as means of 6 measurements per sample.

Syneresis was determined according to Vercet et al. [19]. Approximately 10 g of sample was accurately weighed into a 50 mL pre-weighed centrifuge tube and was centrifuged at 5000× *g* for 5 min expressing the results in % (*w/w*).

### 2.3. Color Analysis

Color analysis was carried out immediately after homogenization of the samples (to prevent color degradation as a result of light and oxygen) using a Minolta Chromameter 2 reflectance colorimeter (Model CM-600 d, Osaka, Japan), equipped with the measurement head CR 300, and D65 source as illuminant. About 20 g of sample was placed in a quartz cell (diameter 35 by 34 mm, Konica Minolta, Osaka, Japan) and each sample was submitted to three replicate measurements. Color was studied in the CIE L*a*b* color space by measuring the values of *L** (lightness), *a** (coordinate red/green) and *b** (coordinate yellow/blue). 

### 2.4. Volatile Compounds 

Volatile compounds were extracted by solid-phase micro-extraction (SPME) and analyzed by a gas-chromatographic system equipped with mass spectrometer (GC-MS) as described in previous papers [20,21]. The volatile compounds were identified by comparison with the mass spectra present in the NIST and Wiley libraries, quantified, and expressed in terms of integrated area.

### 2.5. Statistical Analysis

One-way analysis of variance followed by Tukey’s test, for multiple comparisons, was used to highlight significant differences in pâtés characteristics at a significant level of 5%. Principal component analysis (PCA) was performed on the correlation matrix and used to explore data. All the statistical analyses were conducted by Minitab 17 (Minitab Inc., State College, PA, USA).

## 3. Results and Discussion

### 3.1. Rheological and Color Characteristics 

Table 2 reported hardness, syneresis, and color characteristics of the pâtés under study in relation to the homogenization conditions. Both time and speed significantly influenced the rheological properties of the product. Lopez-Sanchez et al. [11] reported that the rheological properties depend on both the soluble solids in the serum phase and the particles volume fraction of insoluble solids. In particular, the data about the syneresis degree (serum separation of oil and water) showed significantly lower values at longer time and higher speed. Data reflected the behavior observed for texture analysis. This could be due to a more intense and efficient grinding of the ingredients, particularly those with a high degree of fibrousness, which need higher speed and longer times of homogenization. Moreover, higher standard deviations for hardness were observed at lower speed. This result could be due to the higher frequency of particles with non-uniform volume, as a consequence of the non-optimal homogenization phase, thus influencing the repeatability of the results. Higher values of hardness determined a more evident serum separation. Increasing the process time and speed a better homogenization of the ingredients was obtained as well as a softer consistency of the pâtés, preventing serum separation. Syneresis degree negatively influences the consumers quality perception of foods [22]. 

A clear influence of the variables under study on sample color was observed. In particular, the index of lightness (*L**) showed significantly higher values when intermediate time and speed conditions were applied. It is known that the values of *L** are influenced by the degree of mincing and the subsequent incorporation of air bubbles during this phase which determine an increase in the diffusion of light [23]. Furthermore, lightness depends on pH and water holding capacity as well. In our study, mincing did not affect pH and therefore the changes observed in lightness must have been due to other causes, such as the modifications in the structure caused by mincing. The latter, leads to a breakdown of cells making available a greater amount of water on the surface. 

The *a** index showed negative values in all the examined samples, indicating a tendency towards a green color. A general decrease of *a** was obtained increasing process time and speed, likely due to a leakage of chlorophylls from the intercellular space, yielding a more intense bright green color on the vegetable surface. On the other hand, the *b** index, which had positive values, showed an opposing trend, indicating a tendency towards a yellow color when higher speed and longer time were applied. This could be attributed to a greater solubilization of tissue carotenoids by the added oil that occurs when processing parameters caused higher tissues breakdown.

### 3.2. Volatile Compounds 

Table 3 reports the principal chemical classes of the headspace composition, determined by SPME/GC-MS analysis of the samples as a function of time and speed variables. On the whole, the 39 identified volatile compounds were grouped into the following 7 classes: (i) Terpenes (sabinene; D-limonene; eucalyptol; linalool; L-α-terpineol; 1-methoxy-4-(2-propenyl) benzene; β-*cis*-Ocimene; (ii) aldehydes (3-methyl butanal; hexanal; *trans*-2-hexenal; octanal; nonanal; furfural; benzaldehyde; phenylacetaldehyde); (iii) alcohols (ethanol; 1-propanol; 1-penten-3-ol; 3-methyl 1-butanol; *cis*-2-penten-1-ol; 1-hexanol; *cis*-3-hexen-1-ol; *trans*-2-hexen-1-ol; benzyl alcohol; phenyl ethyl alcohol; 1-methyl-2-cyclopenten-1-ol); (iv) ketones (acetone; 2-butanone; 3-hexanone); (v) esters (ethyl acetate; 3-hexen-1-ol acetate; methyl benzoate); (vi) acids (2,2-dimethyl propanoic acid; acetic acid); (vii) others (dipropyl disulfide; octane; piperazine; methyl benzene; ethyl benzene). Terpenes, aldehydes, and alcohols were the most representative volatile compounds, accounting for more than 70% of the total volatile fraction, followed by ketones, esters, acids, and others minor volatiles.

Regarding the time variable, only aldehydes, ketones, and acids were significantly influenced. In particular, aldehydes and acids were significantly increased with longer homogenization time, conversely to ketones, which were significantly decreased. 

Significant influence by the speed variable was observed for all of the volatile compounds. In detail, a significant increase in terpenes and aldehydes occurred with higher homogenization speed and opposingly, alcohols, esters, acids, ketones, and other volatiles significantly decreased. In general, aldehydes with higher molecular weight could act as precursors for volatile alkanals and alkenals for pâtés aroma, although these compounds have less flavor intensity, whereas aldehydes with short chains have sharper flavors. In particular, hexanal, octanal, and nonanal were found to be responsible for green-grassy, green-fresh, and green flavor notes [24]. Dirinck et al. [25] reported that aldehydes should be major contributors to flavors because of their low threshold values and distinct odor characters (for example, rancid, sweet, floral, and pungent notes). The release of ketones is considered the result of greater oxidation of short-chain fatty acids resulting from microbial catabolic pathway in table olives, the main ingredient of pâtés, and characterized by unpleasant, fermented flavor [26]. 

Figure 1 shows the loading plot and score plot of the PCA of the volatile compound classes. The first two components explain over 86% of the total variability. The loading plot shows that PC1, in particular, accounted for 68.5% of variability and was negatively correlated with terpenes and aldehydes, whereas it was positively correlated with alcohols, esters, acids, and other volatiles. Therefore, PC1 allowed us to discriminate samples in the score plot according to homogenization speed and in particular, those obtained by lower speed to the others. Higher homogenization speed corresponded to higher terpenes and aldehydes. The PC2, accounting for about 18% of variability, allowed to separate samples according to the homogenization time whereas those produced in less time showed more ketones. 

## 4. Conclusions

The obtained results showed that obtaining good rheological and sensorial properties in olive-based pâtés is necessary to set up and apply optimal conditions in terms of homogenization time and speed. Indeed, hardness and syneresis were significantly influenced by the increase of both homogenization time and speed. Moreover, *L** values were significantly higher at intermediate time and speed conditions, whereas the volatile fraction was significantly influenced above all by the speed variable. In perspective, the obtained results could be highly useful in olive-based pâtés industries, to save energy consumed by the processing plants and at the same time fulfill the increasing demand of products with agreeable sensory products.

## Figures and Tables

**Figure 1 foods-08-00115-f001:**
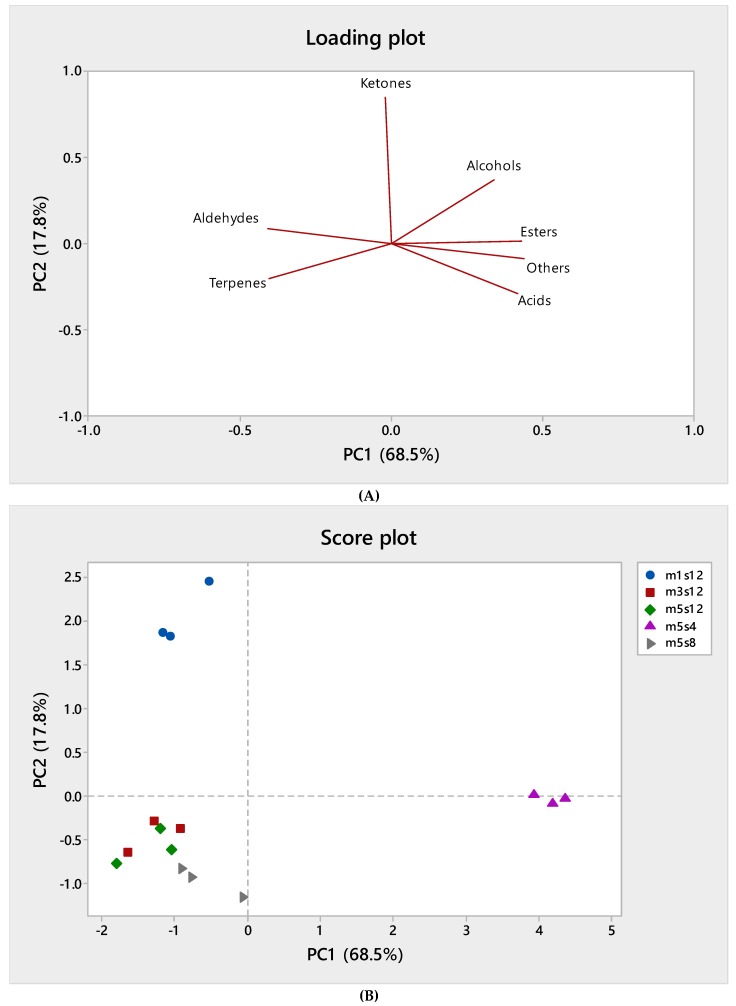
Loading plot (**A**) and score plot (**B**) of the principal component analysis (PCA) carried out on volatiles compounds grouped by chemical classes. For trials coding and description see Table 1.

**Table 1 foods-08-00115-t001:** List of the trials carried out.

Trial	Homogenization Conditions	
	Time (min)	Speed (rpm)
m1s12	1	12,000
m3s12	3	12,000
m5s12	5	12,000
m5s8	5	8000
m5s4	5	4000

**Table 2 foods-08-00115-t002:** Mean values, standard deviation and results of statistical analysis (one-way ANOVA) related to influence of time and speed homogenization variables on texture, syneresis, and color of pâtés under investigation.

Parameter	m1s12	m3s12	m5s12	m5s8	m5s4
Hardness (N)	21.20 ± 0.40 ^a^	17.90 ± 0.50 ^b^	10.90 ± 0.40 ^c,C^	23.00 ± 1.50 ^B^	45.30 ± 3.70 ^A^
Syneresis (% *w/w*)	2.87 ± 0.21 ^a^	2.45 ± 0.08 ^a^	1.73 ± 0.20 ^b,C^	3.55 ± 0.36 ^B^	6.65 ± 0.20 ^A^
*L*	39.08 ± 0.01 ^b^	40.21 ± 0.04 ^a^	38.53 ± 0.03 ^c,B^	39.15 ± 0.01 ^A^	37.73 ± 0.02 ^C^
*a**	−4.48 ± 0.02 ^a^	−4.98 ± 0.03 ^c^	−4.61 ± 0.04 ^b,C^	−4.06 ± 0.02 ^A^	−4.30 ± 0.02 ^B^
*b**	24.13 ± 0.01 ^c^	26.53 ± 0.01 ^b^	27.83 ± 0.04 ^a,A^	26.34 ± 0.03 ^C^	26.77 ± 0.01 ^B^
100 − *L*	60.92 ± 0.01 ^b^	59.80 ± 0.04 ^c^	61.47 ± 0.03 ^a,B^	60.85 ± 0.01 ^C^	62.28 ± 0.02 ^A^

*L*, lightness index; *a**, redness index; *b**, yellow index. a–c and A–C, Lowercase and uppercase different letters indicate significant differences at *p* < 0.05 for time and speed variable, respectively. For trials coding and description see Table 1.

**Table 3 foods-08-00115-t003:** Mean values (integrated area × 10^−6^), standard deviation and results of statistical analysis (one-way ANOVA) related to influence of time and speed homogenization variables on volatile compounds of pâtés under investigation.

Volatile Compounds	m1s12	m3s12	m5s12	m5s8	m5s4
Terpenes	18.00 ± 0.25 ^a^	19.73 ± 1.72 ^a^	19.01 ± 0.97 ^a,A^	21.20 ± 1.58 ^A^	11.62 ± 0.36 ^B^
Aldehydes	40.34 ± 0.86 ^b^	40.18 ± 1.15 ^b^	44.78 ± 1.93 ^a,A^	35.43 ± 2.31 ^B^	26.61 ± 3.33 ^C^
Alcohols	15.19 ± 1.03 ^a^	14.76 ± 0.79 ^a^	14.05 ± 0.53 ^a,B^	13.87 ± 0.23 ^B^	16.49 ± 0.71 ^A^
Ketones	5.73 ± 0.42 ^a^	1.96 ± 0.12 ^b^	2.08 ± 0.08 ^b,B^	2.36 ± 0.03 ^B^	2.73 ± 0.19 ^A^
Esters	4.39 ± 0.38 ^a^	4.29 ± 0.22 ^a^	4.34 ± 0.42 ^a,B^	4.46 ± 0.46 ^B^	6.19 ± 0.38 ^A^
Acids	0.55 ± 0.02 ^c^	0.72 ± 0.05 ^b^	0.80 ± 0.01 ^a,B^	1.22 ± 0.28 ^B^	2.02 ± 0.20 ^A^
Others	6.56 ± 0.27 ^a^	5.45 ± 0.10 ^b^	7.08 ± 0.25 ^a,C^	8.06 ± 0.15 ^B^	13.25 ± 0.40 ^A^

a–c and A–C, lower and uppercase different letters indicate a significant difference at *p* < 0.05 for time and speed variable, respectively. For trials coding and description see Table 1.

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
