# Peer review of "Influence of Homogenization Time and Speed on Rheological and Volatile Composition in Olive-Based Pâtés"

_foods, 2019, doi:10.3390/foods8040115_

Round 1

Reviewer 1 Report

This nicely written paper conveys a clear message on effects of increased homogenization time and speed on the decrease in pâté hardness and syneresis and on the increased content of terpenes and aldehydes in the volatile fraction. And this paper need address the following minor comments on the experimental part for texture analysis.

Could the authors give the dimensions of the beaker (height, diameter) used for textural testing as well as the typical height of pâté samples? The dimensions of beaker relative to plunger and sample height are known to affect the results under certain conditions (see for instance DOI: 10.1016/S0144-8617(98)00113-1

 and  https://doi.org/10.1016/S0144-8617(99)00125-3).

Could the authors explain why the larger the hardness, the larger the relative error on its value? This result is rather counter intuitive, especially if one expects that, for a given sensitivity of the texture analyzer, measurements with harder samples would return more precise hardness.

Author Response

thank you very much for your careful revision and helpful suggestions.

In agreement with your observations, the manuscript has been carefully revised and the following modifications have been introduced.

Reviewers' comments

Could the authors give the dimensions of the beaker (height, diameter) used for textural testing as well as the typical height of pâté samples? The dimensions of beaker relative to plunger and sample height are known to affect the results under certain conditions (see for instance DOI: 10.1016/S0144-8617(98)00113-1

 and  https://doi.org/10.1016/S0144-8617(99)00125-3).

As requested the dimension of the beaker for textural testing as well as the typical height of pâté samples were added (please see page 3, first paragraph of “Rheological analysis” section)

Could the authors explain why the larger the hardness, the larger the relative error on its value? This result is rather counter intuitive, especially if one expects that, for a given sensitivity of the texture analyzer, measurements with harder samples would return more precise hardness.

A possible explanation was added as requested (please see page 4, first paragraph of “3.1. Rheological and colour characteristics”

section)

Moreover, the manuscript was careful revised for English language.

Reviewer 2 Report

This is an interesting paper, though the experimental design is a little limited to five treatments and it might be a little difficult to draw general conclusions from so few a set of trials.

I had some reservations on the presentation e.g. table 3 please do not use "E+05" to denote " x 105" Please modify the table using convetional scientific notation and not excel shorthand.

The level of English was excellent for non-native speakers, but I thought some phrases could be improved e.g. line 32 could read "In recent years...."

Author Response

Thank you very much for your careful revision and helpful suggestions.

In agreement with your observations, the manuscript has been carefully revised and the following modifications have been introduced.

Reviewers' comments

I had some reservations on the presentation e.g. table 3 please do not use "E+05" to denote " x 105" Please modify the table using conventional scientific notation and not excel shorthand.

The table 3 was formatted as requested

The level of English was excellent for non-native speakers, but I thought some phrases could be improved e.g. line 32 could read "In recent years...."

The manuscript was careful revised for English language as requested
